# Spatiotemporal Pattern of Carbon Compensation Potential and Network Association in Urban Agglomerations in the Yellow River Basin

**Haihong Song, Yifan Li * , Liyuan Gu , Jingnan Tang and Xin Zhang**

Urban and Rural Planning, School of Landscape Architecture, Northeast Forestry University,
Harbin 150040, China; sunrising@nefu.edu.cn (H.S.); guliyuan@nefu.edu.cn (L.G.);
tangjingnan@nefu.edu.cn (J.T.); zhangxin1118@nefu.edu.cn (X.Z.)
* Correspondence: liyifan@nefu.edu.cn; Tel.: +86-158-3365-8912

**Abstract:** The Yellow River Basin is an important energy base and economic belt in China, but its water resources are scarce, its ecology is fragile, and the task of achieving the goal of carbon peak and carbon neutrality is arduous. Carbon compensation potential can also be used to study the path to achieving carbon neutrality, which can clarify the potential of one region's carbon sink surplus to be compensated to the other areas. Still, there needs to be more research on the carbon compensation potential of the Yellow River Basin. Therefore, this study calculated the carbon compensation potential using the β convergence test and parameter comparison method. With the help of spatial measurement tools such as GIS, GeoDa, Stata, and social network analysis methods, the spatiotemporal pattern and network structure of the carbon compensation potential in the Yellow River Basin were studied from the perspective of urban agglomeration. The results demonstrate the following: (1) The overall carbon compensation rate of the YRB showed a downward trend from 2005 to 2019, falling by 0.94, and the specific pattern was "high in the northwest and low in the southeast". The spatial distribution is roughly spread along the east–west axis, and the distribution axis and the center of gravity keep shifting to the northwest. It also showed a weak divergence and a bifurcation trend. (2) The carbon compensation rate in the YRB passed the spatial correlation and β convergence tests, demonstrating the existence of spatial correlation and a "catch-up effect" among cities. (3) The overall distribution pattern of the carbon compensation potential in the YRB is a "low in the west and high in the east" pattern, and its value increased by 8.86% during the sampled period. (4) The network correlation of carbon compensation potential in the YRB has been significantly enhanced, with the downstream region being more connected than the upstream region. (5) The Shandong Peninsula Urban Agglomeration has the largest network center, followed by the Central Plains Urban Agglomeration, and the Ningxia along the Yellow River Urban Agglomeration has the fewest linked conduction paths. According to the research results, accurate and efficient planning and development suggestions are proposed for urban agglomeration in the Yellow River Basin.

**Keywords:** carbon compensation potential; spatiotemporal pattern; network association; urban agglomeration; Yellow River basin; β convergence test

## 1. Introduction

The Paris Agreement, adopted by approximately 200 countries and regions in 2015, puts forward the target of "carbon neutrality". China has proposed to achieve carbon peak and carbon neutrality in 2030 and 2060, respectively [1]. The Yellow River Basin (YRB) accounts for more than 50% of China's coal reserves and emits 1.6 times more carbon than the rest of the country [2]. The provinces with the highest coal production in China are Inner Mongolia, Shanxi, and Shaanxi, all part of the YRB. The three provinces also produce the highest carbon emissions in China, with a carbon emission intensity of 6.4, 5.2, and 3.4 times that of the whole country, respectively [2]. The YRB has a large region of arid and

semi-arid regions, most of which are ecologically fragile [3]. Due to the long-term growth of the coal chemical industry, the YRB continues to be indebted to ecological protection beyond the limits of resources, energy, and the environment. Therefore, the carbon effect and carbon compensation potential of the YRB should be scientifically estimated to help achieve carbon neutrality.

In existing research, carbon emissions, carbon sequestration, carbon compensation, and carbon compensation potential have been measured using various methods in various fields. In terms of research on carbon emissions, the involved industries include construction [4], electricity [5], transportation [6], agriculture [7], and plantations [8]. All sectors of life are associated with carbon emissions [9]. The energy supply sector is the sector with the highest carbon emissions [10]. Methods of carbon emissions estimation include measurement using lighting data [11], energy conversion [12], and a life cycle approach [13]. Many scholars believe that nighttime light data can visually reflect the strength of human activity [14]; therefore, it is gradually being used in the study of urban problems. Scholars have found a close relationship between nighttime light (NTL) data and urban carbon dioxide emissions [15]. National Polar-orbiting Partnership-Visible Infrared Imaging Radiometer Suite (NPP-VIIRS) NTL data have a higher resolution than Defense Meteorological Satellite Program's Operational Linescan System (DMSP-OLS) NTL data, and more accurate carbon emission values can be obtained through correction [11]. In terms of carbon sequestration, Tang et al. estimated forest carbon storage based on regional forest resource inventory data, such as forest type and density [16]; with the development of remote sensing and geographic information system technology, some scholars have applied it to estimate the coverage area and biomass of urban vegetation and then calculate carbon sequestration [17]. Carbon sequestration can be estimated using geostatistical modeling to estimate forest biomass at a regional scale, or to quantitatively describe forest carbon cycling processes and estimate forest carbon stocks [18]. In a study of carbon dioxide emissions and storage in counties in China, Chen et al. used MODIS NPP products to obtain net primary productivity data and then calculated the carbon sequestration value of land vegetation based on the conversion factor (1.62/0.45) [19]. As for the measurement of carbon compensation, Wu et al. defines the ratio of carbon absorption to carbon emissions as the carbon compensation rate and studied the evolution and distribution characteristics of the carbon compensation rate in China's plantation industry [8], He et al. defined the value of carbon absorption and carbon emission reduction as the carbon offset value [20]. Yan et al. used carbon deficit sensitivity and ecological compensation factors to study the quantification of ecological compensation standards [21]. In terms of carbon compensation potential, Zhou et al. used the entropy weight method to calculate the carbon emission reduction potential index and graded the carbon emission reduction potential from the county level [22]. Wu et al. verified the convergence of China's agricultural carbon compensation rate using spatially correlated and β convergence methods and calculated carbon compensation potential using a parameter comparison method [7]. He et al. believes that carbon offset potential takes into account the dual effects of carbon sequestration and carbon emissions and that carbon reduction potential is more limited in comparison, and used absolute convergence and conditional convergence to measure China's carbon offset potential [20].

Researchers have further combined spatiotemporal pattern methods to conduct research, involving fields such as characterizing spatiotemporal patterns, evolutionary features, network correlations, etc., and mostly used spatial econometric methods such as Moran's I, kernel density estimation, and cold and hot spot analysis [13]. Li et al. studied the spatial relevance and spatial gathering characteristics of carbon emissions in the Jing-Jin-Ji and depicted the spatial and temporal pattern in combination with time series [23]. He et al. summarized the evolution law of carbon productivity by studying the geographical features of the region, and used the nuclear density curve to fit the dynamic evolution process of carbon productivity in China [20]. Song et al. investigated the carbon emission correlation of the Chengdu Chongqing city cluster and analyzed the overall network

structure, agglomerated subgroup correlation characteristics, and individual centrality in the network [24]. Ren et al. used the life cycle method to calculate the carbon emissions of China's construction industry and analyze the correlation evolution characteristics of carbon emission networks [13]. Zhang et al. analyzed the network correlation structure of carbon emissions in Beijing-Tianjin-Hebei and constructed a carbon-neutral zone [25]. Li et al. found that the spatial distribution characteristics of carbon emissions and carbon absorption were obvious, and this feature was used to divide Wuhan urban area into carbon compensation zones, including carbon compensation payment area replenishment area and balance area [26].

Currently, the sense of distance between cities is gradually disappearing, and agglomeration effects such as resources, capital, and talent are generated to form urban agglomerations. The formation of urban agglomerations can promote communication and cooperation between cities, improving economic efficiency and the social development level. Chen's research shows that the industrial composition of urban agglomerations plays a substantial role in $CO_2$ emission and influences the intensity and spatial distribution of carbon emission [27]. Zhao et al. found that China's urban agglomerations have obvious networking trends, and the cyberspatial structure of urban agglomerations has significant emission reduction effects [28]. The core cities in the urban agglomeration of the YRB play an important role and have a certain radiation driving effect on the surrounding cities [1]. In China's 14th Five-Year Plan, it is emphasized that overall development efficiency should be driven by areas with advantages, such as central cities and urban agglomerations, so more attention should be paid to the spatial model of urban agglomerations in the study of urban development [29]. In general, studying the carbon compensation potential from the perspective of urban agglomerations is of great significance to promoting coordinated development and cooperation among cities, thereby achieving carbon peak and carbon neutrality.

Based on the existing literature, carbon compensation can reflect the regional carbon budget and expenditure more intuitively, and the accounting methods and spatial distribution of carbon compensation and carbon compensation potential are relatively rich, but the research on carbon compensation potential and its network correlation in urban agglomerations is weak. The following are innovative points in this study regarding research on carbon compensation potential: (1) Urban agglomerations were selected as the research module for carbon compensation potential, breaking the administrative boundaries, and dividing urban agglomerations according to socioeconomic and other development conditions. This study examined not only whole urban agglomerations but also typical urban agglomerations individually to determine the characteristics of potential urban carbon compensation potential. (2) The social network analysis (SNA) method was introduced to the research of spatial and temporal patterns of urban carbon compensation potential to explore the spatial characteristics of the carbon compensation potential network and the association mechanism in a hierarchical manner. Based on the existing literature research results and the development of the YRB, the following research objectives were formulated for this study: (1) To analyze the spatiotemporal patterns and evolution characteristics of carbon compensation rates in urban agglomerations in the YRB and summarize the changes in the overall carbon budget in time and space. (2) To examine the spatial convergence characteristics of carbon compensation rate and calculate the carbon compensation potential in order to visually represent the difference in carbon budget and expenditure between regions. (3) To analyze the overall network structure characteristics of carbon compensation potential in the YRB, the internal correlation characteristics of urban agglomerations with strong correlations, and the distribution of individual cities with high centrality index. (4) Based on the research results, propose planning and development suggestions for the YRB in the context of carbon neutrality.

## 2. Study Area and Data

### 2.1. Study Area

In this study, the sample was 7 major urban agglomerations in the YRB, including 56 cities in 9 provinces. The specific names of the cities are shown in Table 1. The total land area of the study area was approximately 772,284.6 km². The spatiotemporal pattern and network spatial association of carbon compensation potential in this region from 2005 to 2019 were studied. Figure 1 shows the schematic diagram of the study area. Among them, the Lanxi Urban Agglomeration and Ningxia Cities along the Yellow River Group are located in the upper reaches of the YRB. In contrast, the Hubao-Eyu Urban Agglomeration, Jinzhong Urban Agglomeration, and Guanzhong Plain Urban Agglomeration are situated in the middle reaches of the Yellow River, while the Central Plains Urban Agglomeration and Shandong Peninsula Urban Agglomeration are located in the lower reaches of the Yellow River. The Central Plains Urban Agglomeration and Guanzhong Plain Urban Agglomeration of the study area are world-class urban agglomerations in China, and the Lanxi Urban Agglomeration and Hubao-Eyu Urban Agglomeration are national urban agglomerations.

**Table 1.** Name of cities in the study area.

| Urban Agglomeration | City | Urban Agglomeration | City |
|---|---|---|---|
| Central Plains Urban Agglomeration | Hebi Luohe Jiaozuo Kaifeng Luoyang | Lanxi Urban Agglomeration | Baiyin Dingxi Lanzhou Tianshui Xining |
| | Pingdingshan Shangqiu Xinxiang Xuchang | Ningxia Cities along the Yellow River Group | Shizuishan Wuzhong Yinchuan Zhongwei |
| | Zhengzhoou Zhoukou Jincheng Changzhi Bozhou | Hubao-Eyu Urban Agglomeration | Baotou Bayan Nur Ordos Hohhot Yulin |
| Shandong Peninsula Urban Agglomeration | Binzhou Dezhoou Dongying Heze Jinan Jining | Guanzhong Plain Urban Agglomeration | Tongchuan Baoji Shangluo Weinan Xi'an Xianyang |
| | Liaocheng Linyi Qingdao Rizhao Taian Weihai Weifang Yantai Zaozhuang Zibo | Jinzhong Urban Agglomeration | Linfen Lvliang Jinzhong Taiyuan Xinzhou Yangquan |

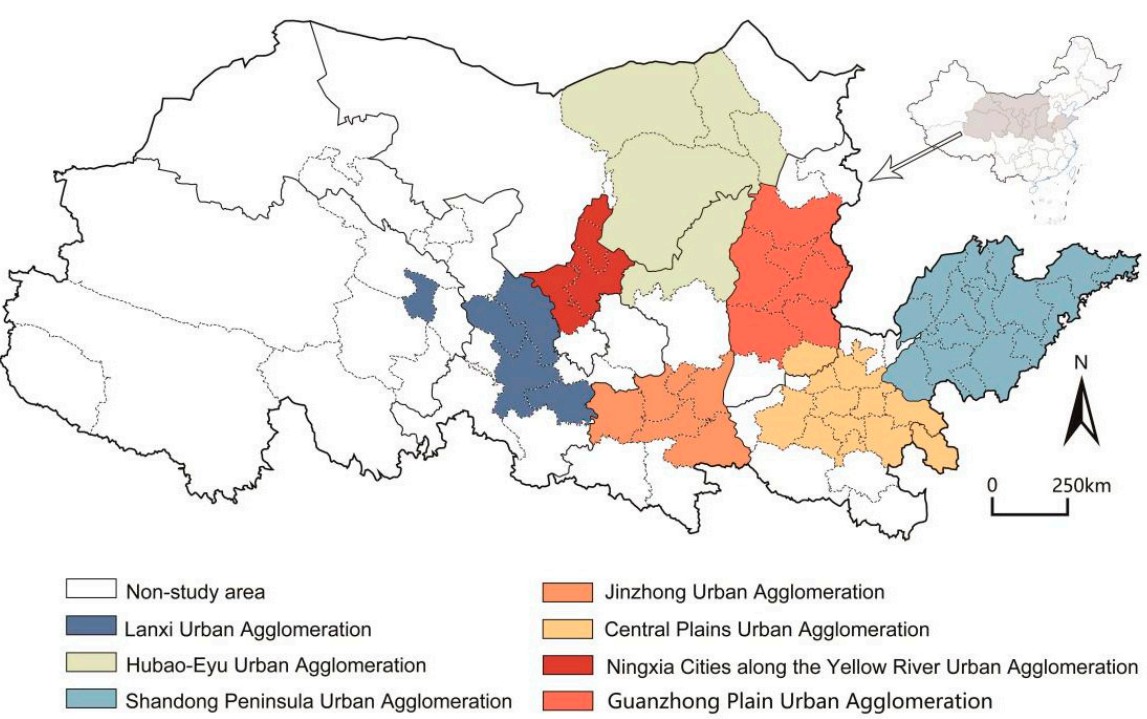

Note: This drawing is based on the standard map of the Ministry of Natural Resources of The People's Republic of China, Drawing review number: GS(2022)4309.

**Figure 1.** Study area.

### 2.2. Data Sources

The data are divided into four major sections: (1) Map data were obtained from the standard map service of the Ministry of Natural Resources of China (bzdt.ch.mnr.gov.cn accessed on 7 November 2022), and all maps in the paper were drawn based on the map with the review number GS (2022) 4309. (2) The carbon sequestration data were calculated from the initial data of the MODIS platform (https://modis.gsfc.nasa.gov/ accessed on 14 October 2022), which is reflected in 3.1. (3) Carbon emission data were retrieved from the CEDAs database (www.ceads.net accessed on 14 October 2022) [30]. (4) The socioeconomic data were obtained from the China Statistical Yearbook. The data obtained had missing data in some years for individual cities, and they were all interpolated to complete the data.

## 3. Methodology

### 3.1. Carbon Sequestration Value Calculation Method

The steps to obtain carbon sequestration data are to first calculate the average annual NPP of each city in the YRB according to the MOD17A3 data provided by the MODIS platform, then calculate the net primary productivity of each city from 2005 to 2019 according to the area, and finally convert the NPP data to carbon sequestration according to the conversion factor in vegetation: NPP = (NPP 0.45) × 1.62 [19].

### 3.2. Carbon Compensation Rate Calculation Method

The carbon compensation rate is an indicator used to gauge the carbon balance of an area, and the carbon compensation rate is the ratio of the carbon sequestration value to the carbon emissions in the area [8]. *CCR* (carbon compensation rate) is calculated as follows:

$$CCR = CSV/CE \tag{1}$$

where *CSV* represents the amount of carbon sequestered by the city and *CE* represents the carbon emissions. If *CCR* > 1, the amount of carbon sequestered in the city exceeds the amount emitted, and it belongs to the carbon sink area in the urban ecosystem; if *CCR* < 1

is greater than the amount of carbon sequestered, it belongs to the carbon source area in the urban ecosystem; and if *CCR* = 1, the city achieves carbon balance.

### 3.3. Carbon Compensation Potential Calculation Method

(1) Spatial correlation test

In this study, global Moran's I [31] auto correlation was used to examine whether there was a spatial correlation in carbon compensation rates among cities, using the inverse distance weight model and global auto correlation analysis with the help of GeoDa software. The formula for global autocorrelation is:

$$I = \frac{n \cdot \sum_{i=1}^{n} \sum_{j=1}^{n} w_{ij} p_i p_j}{\sum_{i=1}^{n} \sum_{j=1}^{n} w_{ij} \cdot \sum_{i=1}^{n} p_i^2} \tag{2}$$

where *I* is Moran's index, *n* is the number of observations, $W_{ij}$ is the inverse distance space weights matrix of points *i* and *j*, and *p* is the amount that each point is different from the mean.

(2) β convergence test

The concept of β convergence originated from studies in economics to verify the convergence of income levels and economic growth rates [32]. The β convergence referred to in this study is that the low-value region of carbon compensation potential is catching up with the carbon compensation value of the high-value region at a higher growth rate than the high-value area. Furthermore, the disparity between the high-value region and the low-value region decreases over time. Conditional β convergence is based on absolute β convergence by adding variables so that the convergence results fit the study more closely. At present, there is a lack of literature to study the influencing factors of urban carbon compensation rate, so this study selects the influencing factors that may be related to urban carbon compensation rate by referring to the influencing factors of agricultural carbon compensation rate, plantation carbon compensation rate, and urban carbon emissions and combined with the research of Wu et al., Li et al., Nie et al., Zhang et al., Cui et al., five control variables of GDP [33,34], scale of the population [35], industrial structure [31], urbanization rate [35], and carbon productivity [7] were selected for conditional beta convergence analysis. In the subsequent investigation, it was verified that these variables correlate with the carbon compensation rate.

The absolute β convergence model [8] is represented as follows:

$$\ln(\frac{y_{i,t+1}}{y_{i,t}}) = \alpha + \beta \ln y_{i,t} + \mu_i + \eta_t + \varepsilon_{i,t} \tag{3}$$

where $y_{i,t}$ denotes the carbon compensation rate of the city *i* in period *t*, and it represents the carbon compensation rate of the city *i* in period *t* + 1; $\alpha$ is a constant; $\mu_i$ and $\eta_t$ represent the area effect and time effect; and $\varepsilon_{i,t}$ is the error term. If β < 0, it indicates that there is convergence and a catch-up effect from the area with a low carbon compensation rate to the area with a high value. The convergence rate is given by—*ln* (1 + *β*)/*T*. If it is greater than zero, there is divergence.

The conditional β convergence model is:

$$\ln\left(\frac{y_{i,t+1}}{y_{i,t}}\right) = \alpha + \beta \ln y_{i,t} + \lambda \sum_{j=1}^{n} \ln \text{Control}_{i,t} + \mu_i + \eta_t + \varepsilon_{i,t} \tag{4}$$

where $\lambda$ is the control variable parameter, and $Control_{i,t}$ denotes the variables affecting the urban carbon compensation rate; *j* represents the jth variable; and the other symbols have the same meaning as in Equation (3).

(3) Calculation of carbon compensation potential

The results of the β convergence test of the carbon compensation rate values of the cities in the YRB showed that there is convergence. Combined with data from the literature, this study used the parametric comparison method [36] to calculate the carbon compensation potential. The expression of carbon compensation potential is:

$$CCP = 1 - \frac{CCR_i}{CCR_{max}} \tag{5}$$

where *CCP* represents the carbon compensation potential of a city, which takes values in the range of (−1, 1). $CCR_i$ represents the carbon compensation rate of a city, and $CCR_{max}$ represents the maximum value of the carbon compensation rate of each city in the YRB. The larger the value, the greater the potential of the city's carbon sink to compensate for other cities, and vice versa, the smaller the value, the smaller the potential to be tapped and the larger the city's carbon sink.

### 3.4. Spatio-Temporal Pattern Research Methods

(1)　SDE

Using the spatial statistics tools of ArcGIS 10.6 to characterize the distribution of a city's carbon compensation potential. In this study, changes in ellipse area, length of major and minor axes, azimuth angle, and center position are used to show the change in distribution characteristics [20]. The ellipse azimuth and center position were calculated as:

$$\tan\theta = \frac{\left(\sum_{i=1}^n w_i^2 \widetilde{x}_i^2 - \sum_{i=1}^n w_i^2 \widetilde{y}_i^2\right) + \sqrt{\left(\sum_{i=1}^n w_i^2 \widetilde{x}_i^2 - \sum_{i=1}^n w_i^2 \widetilde{y}_i^2\right)^2 + 4\sum_{i=1}^n w_i^2 \widetilde{x}_i^2 \widetilde{y}_i^2}}{2a} \tag{6}$$

$$\overline{x}_w = \frac{\sum_{i=1}^n w_i x_i}{\sum_{i=1}^n w_i} \quad \overline{y}_w = \frac{\sum_{i=1}^n w_i y_i}{\sum_{i=1}^n w_i} \tag{7}$$

where θ is the azimuthal angle of the ellipse, $x_i$ and $y_i$ denote the spatial position of the study object, $\widetilde{x}_i$ and $\widetilde{y}_i$ are the deviation of the object from the mean center, $w_i$ represents the weights, and $x_w$ and $y_w$ are the mean center positions.

(2)　Local spatial autocorrelation

In this study, local spatial autocorrelation was used to investigate the temporal and spatial evolution of urban carbon compensation rate and carbon compensation potential. According to the degree of spatial homogeneity, the study area is divided into high–high and low–low clustering as well as high–low and low–high clustering. The local spatial auto correlation [37] was calculated as:

$$I_i = \frac{n(p_i - \overline{p})}{\sum_{j=1, j \neq i}^n (p_i - \overline{p})^2} \sum_{j=1, j \neq i}^n w_{ij}\left(p_j - \overline{p}\right) \tag{8}$$

where $I_i$ is the local Moran's *I*; $w_{ij}$ is the spatial inverse distance weight matrix of points *i* and *j*, *n* is the number of observations, and p denotes the deviation of individuals from the mean. The larger the $I_i$, the higher the homogeneity of the measured area.

(3)　Kernel density estimation

Kernel density estimation is mostly used to study unbalanced spatial characteristics. It works by fitting sample data to a peak function and using a continuous curve to show the distribution pattern of variables [32]. By observing the position and extension pattern of the kernel density curve, its evolutionary characteristics are summarized. The density function of random variables is:

$$f(x) = \frac{1}{Nh} \sum_{i=1}^{N} K\left(\frac{X_i - \bar{x}}{h}\right) \tag{9}$$

where $N$ is the number of observations, and $X_i$ is the observed value of an identical independent distribution; $\bar{x}$ is the mean value; $K$ is the kernel density; $h$ represents the bandwidth; and $h$ is proportional to the smoothness of the curve and inversely proportional to the accuracy.

*3.5. Methods of Network Association Structure*

(1) Gravitational model

In this study, the correlation characteristics of carbon compensation potential networks between cities were analyzed based on the gravity model. The calculated equation of the original gravity model was as follows:

$$F = G\frac{M_1 M_2}{d^2} \tag{10}$$

In this study, population size, GDP, carbon offset potential, and geographic distance between cities were added to the original gravity model to measure the strength of the association. The modified gravity model [38] was calculated as follows:

$$F_{ij} = k_{ij} \cdot \frac{\sqrt[3]{G_i P_i O_i} \cdot \sqrt[3]{G_j P_j O_j}}{d_{ij}^2}, k = \frac{O_i}{O_i + O_j} \tag{11}$$

where $i$ and $j$ represent city $i$ and city $j$, $F_{ij}$ denotes the gravitational value of the carbon compensation potential between two cities; $k_{ij}$ denotes the adjustment factor of carbon compensation potential between two cities, $G$ represents the city GDP; $P$ represents the city population size; and $O$ represents the carbon compensation potential of cities. The Equation (9) results form a gravitational matrix with $i$ rows and $j$ columns of gravitational values between cities in the YRB. In order to facilitate subsequent research, the gravitational matrix was binarized, and if the gravitational value between the two cities was greater than the average, the value was 1, and for the opposite the value was 0.

(2) Cohesive subgroup analysis

Cohesive subgroups are subsets of actors with direct, close, and positive relationships, and urban network cohesion subgroups can analyze the state of urban agglomerations and internal substructures [39]. The CONCOR program in Ucinet analyzes the correlation coefficient matrix and iteratively calculates the correlation coefficient of each row and column of the matrix many times, measures the similarity between cities, and finally uses the tree diagram to express the grouping and the members within the group [39].

Using UCINET6 software, three typical urban agglomerations in YRB were analyzed as CONCOR cohesive subgroups. Through the parameter settings (maximum segmentation depth to 2, the convergence criterion of 0.2, and maximum iteration of 25), the software divides the city agglomeration into four groups [40] and summarizes them into two-way spillover plate, net spillover sector, net income sector and broker sector according to the correlation characteristics of each group [41].

To analyze network association characteristics of the cohesive subgroups, four measures of network density, network connectedness, network hierarchy, and network efficiency were introduced [24]. Table 2 shows the relevant network metrics.

**Table 2.** The relevant network metrics.

| Index | Format | Symbol Meaning | Function |
|---|---|---|---|
| Network density | $D = \frac{L}{\frac{N(N-1)}{2}}$ | Where $L$ is the number of connections as a whole, and $N$ is the number of points that create links. | Network density is a measure of the overall linkage strength, using the ratio of the number of actual links to the number of possible links, taking a value of 0–1; larger values indicate a more vital overall linkage. |
| Network connectedness | $C = 1 - \frac{V}{\frac{N(N-1)}{2}}$ | Where $V$ is the number of points that are far away from each other and can not be reached from each other, and $N$ is the total number of points. | Network connectedness was used as a measure of overall connectivity, and a higher value indicated fewer isolated points in the overall network. |
| Network hierarchy | $H = 1 - \frac{R}{R_{max}}$ | Where $R$ represents the number of symmetrically accessible points in the network, and $R_{max}$ denotes the total number of symmetric points. | The network hierarchy indicates the indicator of asymmetric reachability between two bodies, and more unidirectional links in the network indicate a tighter network structure. |
| Network efficiency | $E = 1 - \frac{M}{M_{max}}$ | Where $M$ represents the number of redundant lines in the network, and $M_{max}$ denotes the number of all possible strings. | Network efficiency is an indicator used to measure the degree of line redundancy in the network; the lower the number of redundant lines, the higher the network efficiency. |

(3) Individual Analysis Metrics

To analyze individual characteristics in the network, degree centrality, closeness centrality, and between-ness centrality metrics were used [41]. Degree centrality includes point-in degree centrality and point-out degree centrality, which were used to measure the attractiveness and linkage of individuals to other cities, respectively, and are expressed in terms of the number of connections that occurred. The point-in and point-out degrees of centrality were calculated as follows:

$$C_{od}(i) = \frac{\sum_{j=1}^{n} X_{ij}}{n-1} \tag{12}$$

$$C_{id}(i) = \frac{\sum_{i=1}^{n} X_{ji}}{n-1} \tag{13}$$

where $i$ and $j$ denote city $i$ and city $j$, X indicates the presence or absence of ties between two individuals, and $X$ is 1 if ties exist and 0 if they do not.

Closeness centrality includes inward point-in closeness centrality and point-out centrality, which were also used to measure the strength of the connection between individuals and other cities and were expressed as the sum of the shortest distances between individuals. The point-in closeness centrality and point-out centrality were calculated as follows:

$$C_{oc}(i) = \frac{n-1}{\sum_{j=1}^{n} d(i,j)} \tag{14}$$

$$C_{ic}(i) = \frac{n-1}{\sum_{j=1}^{n} d(j,i)} \tag{15}$$

where $d$ denotes the length of the shortest distance between city $i$ and city $j$. If a point has a high proximity centrality, the other points have less control over the point.

The between-ness centrality was used to measure the extent to which a point acts as an intermediate point connecting other points and was expressed in terms of the number of shortest paths in which the issue was located. If the intermediary centrality of a point is

high, it has a strong ability to control information transmission and has a great influence on the network. The formula for between-ness centrality is:

$$C_b(i) = \frac{\sum_{j<k} \frac{g_{jk}(i)}{g_{jk}}}{(n-1)(n-2)} \qquad (16)$$

where $g_{jk}$ is the total number of shortest paths from city $j$ to city $k$ that generate connections, $g_{jk}(i)$ is the shortest paths through the city $i$ for city $j$ to city $k$ to generate connections.

## 4. Results

### 4.1. Spatial and Temporal Patterns of Carbon Compensation Rates in Urban Agglomerations in the YRB

#### 4.1.1. Trends in the Carbon Compensation Rate over Time

Figure 2 shows the change in carbon compensation rates in the YRB and its seven urban agglomerations between 2005 and 2019. The overall carbon compensation rate in the YRB showed a steady downward trend. In 2019, the overall carbon compensation rate decreased by approximately 0.94 compared to 2005. This result shows that the growth rate of overall carbon sequestration is smaller than the growth rate of carbon emissions in the YRB. Among them, the Central Plains Urban Agglomeration, the Shandong Peninsula Urban Agglomeration, and the Ningxia Cities along the Yellow River Group exhibited fewer changes than other urban agglomerations. The carbon compensation rate fluctuated greatly, but the largest change was in the Guanzhong Plain Urban Agglomeration, which decreased by 2.61 during the study period. In addition, the results show that the carbon compensation rate of the Lanxi Urban Agglomeration has an upward trend, increasing by approximately 1.28 in 2019 compared with 2005, indicating that the growth rate of carbon sequestration in this region was greater than the growth rate of carbon emissions.

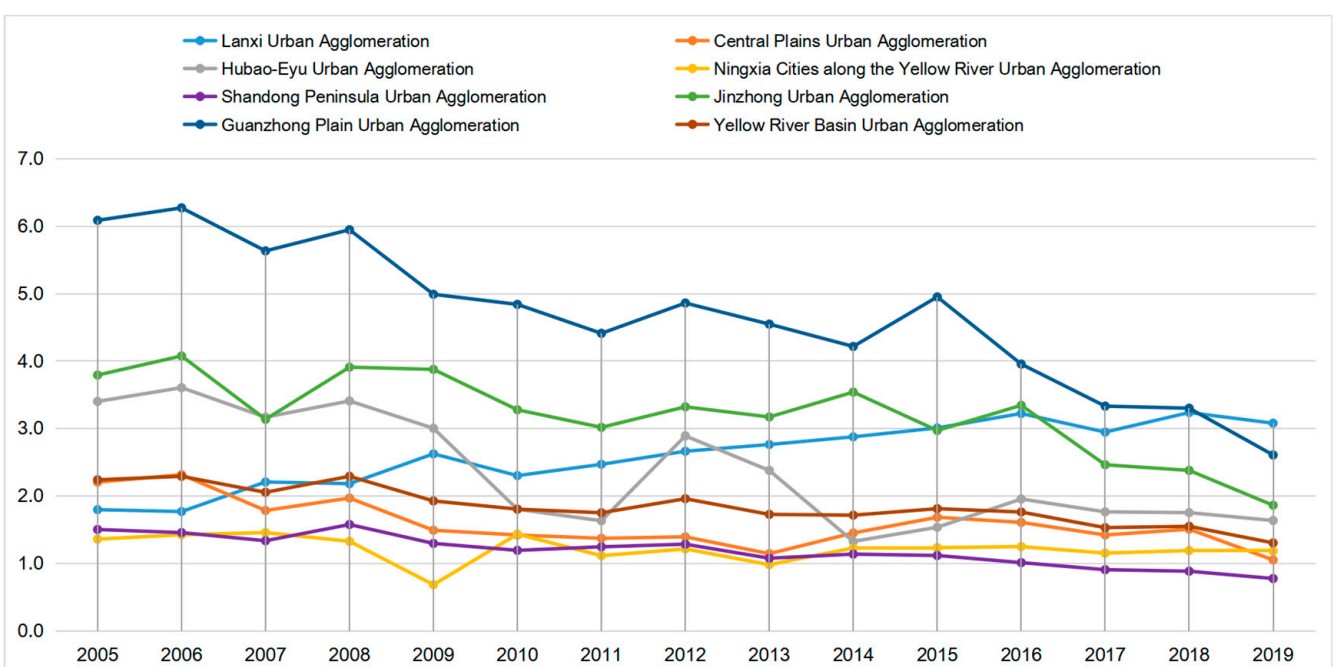

**Figure 2.** Carbon compensation rate of the YRB urban agglomeration.

#### 4.1.2. Spatial Characteristics of the Carbon Compensation Rate

Figure 3 shows the visualization of carbon compensation rate values of four years, 2005, 2009, 2014, and 2019, for the YRB urban agglomerations using the five-level natural breakpoint method. Darker colors represent higher carbon compensation rates. From

Figure 3, it is easy to see that the overall spatial pattern of the carbon compensation rate in the study area varied less in the west and more in the east. The northern part of the Hubao-Eyu Urban Agglomeration, the southwest of the Guanzhong Plain Urban Agglomeration, the northwestern part of the Jinzhong Plain Urban Agglomeration, and the southeast of the Central Plains Urban Agglomeration were areas with high carbon compensation rates. In contrast, the other areas had a relatively low carbon compensation rate. The largest difference in carbon compensation rate within the urban agglomeration was the Hubao-Eyu Urban Agglomeration. In Figure 3, the color of the Shandong Peninsula Urban Agglomeration becomes lighter with time, which indicates that the carbon compensation rate is decreasing year by year and the carbon emission rate is increasing. The area with the slightest difference in carbon compensation rate within the urban agglomeration was the Shandong Peninsula Urban Agglomeration. The carbon compensation rate of its central cities, such as Jinan, Zibo, and Dongying, was smaller compared with other cities in the urban agglomeration. The urban agglomerations in the YRB have different location and resource endowment conditions, so their economic development and ecological environment status are quite different. Generally speaking, the spatial pattern of the carbon compensation rate of urban agglomerations in the YRB showed a pattern of "high in the west and low in the east". The development trend in the west is relatively stable, and the downward trend in the east is obvious.

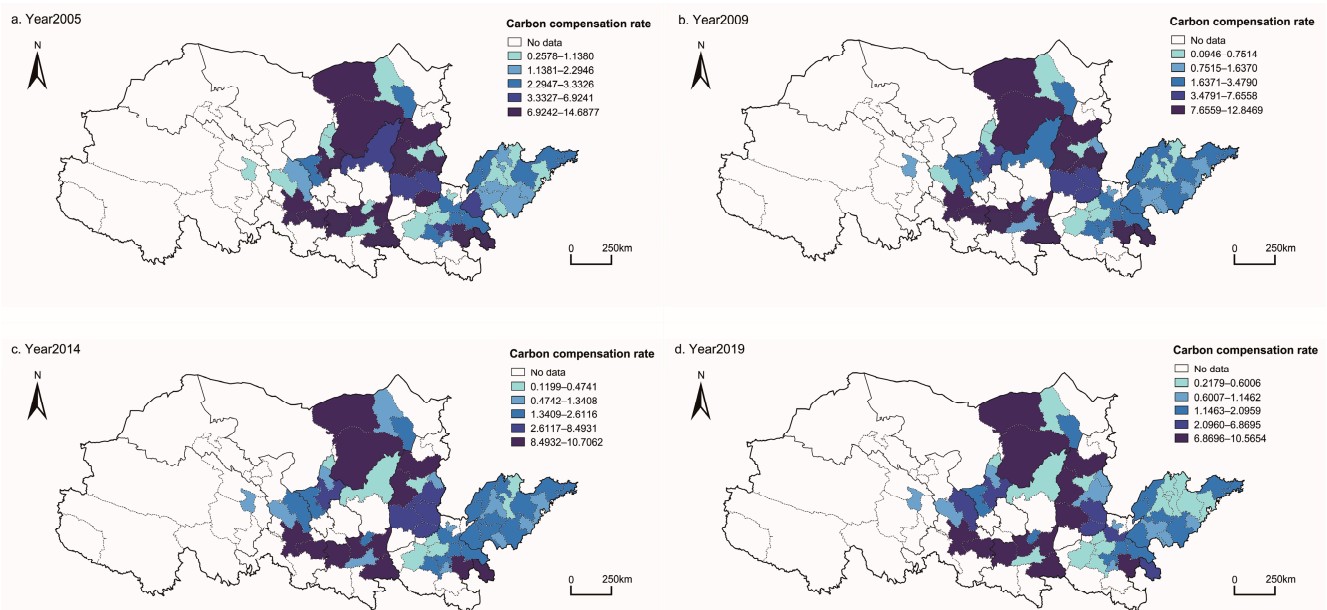

**Figure 3.** Spatial distribution of carbon compensation rate in the YRB urban agglomeration: (**a**) in 2005; (**b**) in 2009; (**c**) in 2014; (**d**) in 2019.

*4.2. Spatial and Temporal Evolution Characteristics of Carbon Compensation Rate in Urban Agglomerations in the YRB*

4.2.1. Migration Characteristics of the Spatial Center of the Carbon Compensation Rate

Figure 4 shows the spatial distribution correlation properties and center of gravity shift trajectories of carbon compensation rate values in YRB urban agglomerations in 2005, 2009, 2014 and 2019. The ellipse in the figure indicates the location of the carbon compensation rate roughly concentrated, and the elliptical long axis is distributed in the east–west direction.

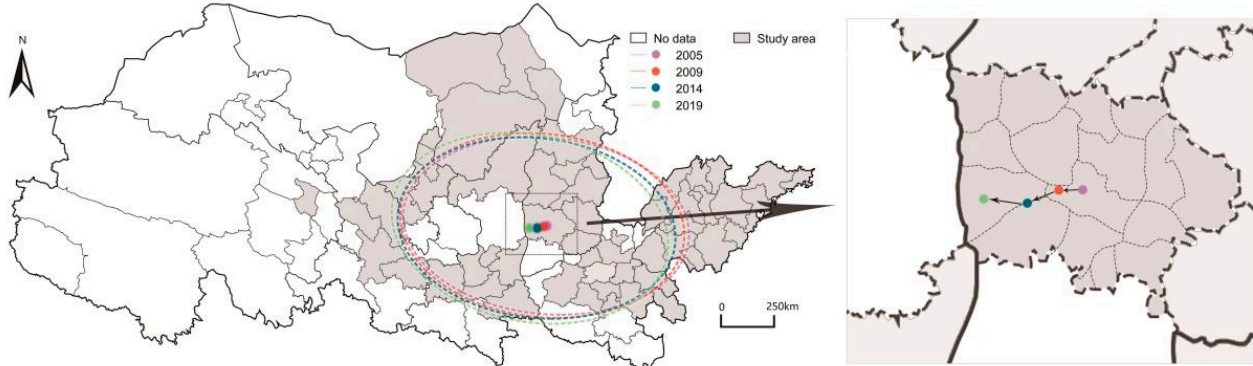

**Figure 4.** Evolutionary characteristics of the spatial distribution of carbon compensation rate in the YRB.

The position of the ellipse has been moving westward as time advances, and the most extended length of the ellipse shortened from 31.59 km to 30.66 km in 2005. This indicates that the carbon compensation rate of the YRB urban agglomerations shows a trend of development toward the center. The area of the ellipse continues to shrink: 53.41 km$^2$ in 2005 and 51.49 km$^2$ in 2019. Both the long and short axes of the ellipse are shortening, from 7.07 km and 2.41 km in 2005 to 6.82 km and 2.40 km, respectively. Moreover, it can be seen that the change is more pronounced in the long than the short axis, which indicates that the spatial pattern of the carbon compensation rate was more variable in the east–west direction and shows a more clustered development. The turning angle of the ellipse was 93.94 in 2005 and 94.15 in 2019, the distribution axis shifted 0.21° to the northwest. The long axis of the ellipse moved southeast–northwest, indicating that the carbon compensation rate changed more significantly in the west, and the decline rate of the carbon compensation rate in the east was greater than the average in the study area. The center of the ellipse was within the Jinzhong Urban Agglomeration from 2005 to 2019, located explicitly in Linfen City, Shanxi. It kept migrating westward over time, along the migration path of Yaodu District–Xiangning County–Ji County. In general, the central migration and spatiotemporal changes in the carbon compensation rate in the YRB were small, and the development was relatively stable.

4.2.2. Dynamic Evolution Characteristics of Kernel Density for Carbon Compensation Rate

Figure 5 shows the kernel density curve of the carbon compensation rate values of the YRB and seven major urban agglomerations over time series. The center and interval of the curve of the overall carbon compensation rate in the YRB showed a leftward shift, indicating a general decrease in the carbon compensation rate. The prominent peak of the curve increased and narrowed, indicating an increasing trend in the agglomeration of carbon compensation rate distribution. The curve had side peaks and small side peaks, indicating that the overall carbon compensation rate in the YRB had a two-stage differentiation trend; however, the trend was weak.

The curve distribution patterns of the seven urban agglomerations were different. The distribution pattern of the curve of the Central Plains Urban Agglomeration showed an agglomeration trend, and the polarization trend has been gradually weakening. The nuclear density curve of the Lanxi Urban Agglomeration showed a rightward shift phenomenon. Furthermore, the main peak was not obvious, and the side peak was close to the main peak, indicating that the carbon compensation rate has been increasing and the distribution dispersion degree was high. The curve of Ningxia Cities along the Yellow River Group varies greatly, at first the main peak was not obvious, the degree of dispersion was high. Then, the curve gradually peaked, and the curve width narrowed, and the degree of agglomeration increased significantly. There was a leftward shift in the curve of the Guanzhong Urban Agglomeration, and the main peak was slightly expanded, indicating

that its carbon compensation rate had decreased and there was a weak discrete trend. The peak of the curve of Hubao-Eyu Urban Agglomeration continued to move left with time and decreased, the width increased slightly, and the peak gap between the main peak and the side peak gradually narrowed. This indicates that the carbon compensation rate of Hubao-Eyu Urban Agglomeration continued to decrease with a discrete trend during the sampled period, and the gradient effect continued to decrease. The Jinzhong Urban Agglomeration had a bipolar or multi-polar differentiation trend. The curve of the Shandong Peninsula Urban Agglomeration shifted significantly to the left, the difference between the main peak and the side peak was not large, there were two poles and the gradient effect was not obvious. In addition, the curve width was significantly narrowed and the degree of agglomeration increased significantly. In general, the evolution characteristics of carbon compensation rates differed, which is related to the large gap between the development conditions in this region and development conditions of urban agglomerations.

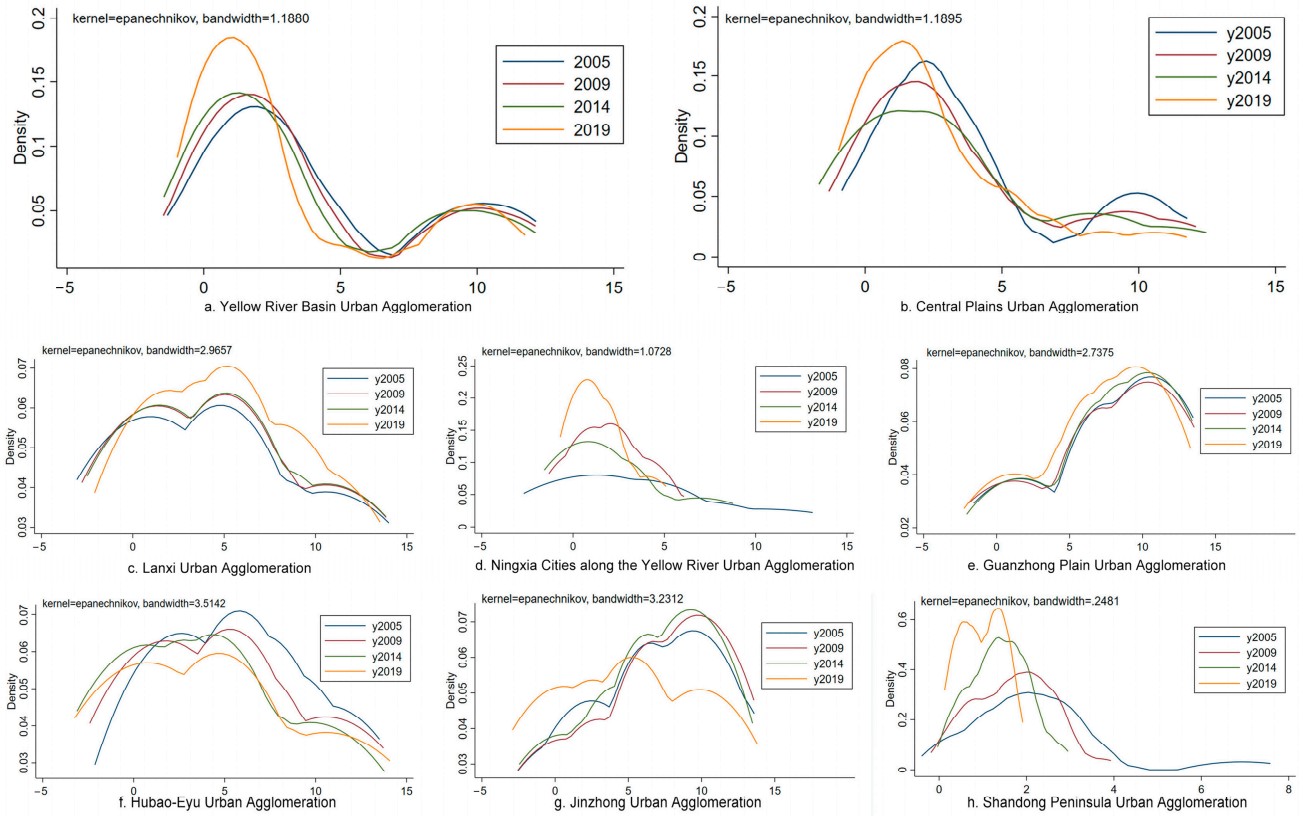

**Figure 5.** Carbon compensation rate evolution characteristics with time series for urban clusters in the YRB.

### 4.3. Spatial Correlation Test of the Carbon Compensation Rate of Urban Agglomerations in the YRB

#### 4.3.1. Carbon Compensation Rate Global Correlation

The global correlation analysis of carbon compensation rate values in the YRB for 2005, 2009, 2014, and 2019 was performed using ArcGIS 10.6 software. The adjacency matrix was discarded using inverse distance matrix analysis to consider the effect of distance on the results of the spatial correlation analysis. The value for Moran's I index of the study area was 0.4 in 2005, showing a high level of aggregation. Furthermore, it was 0.123 and 0.124 in 2009 and 2014, respectively, with little change in the index during this period. The index value in 2019 is 0.112, which was lower compared to the other three years, corresponding to a weak divergence pattern.

Figure 6 shows the LISA clustering plots obtained by analyzing the carbon compensation rate of the YRB for 2005, 2009, 2014, and 2019 using GeoDa software. Compared with the cluster maps for the four years, we found that the 2005 clustering pattern showed a greater difference compared with the other three years. The cluster map for 2005 showed a higher clustering pattern, with subsequent years being more dispersed compared to 2005 and with less change in the clustering pattern.

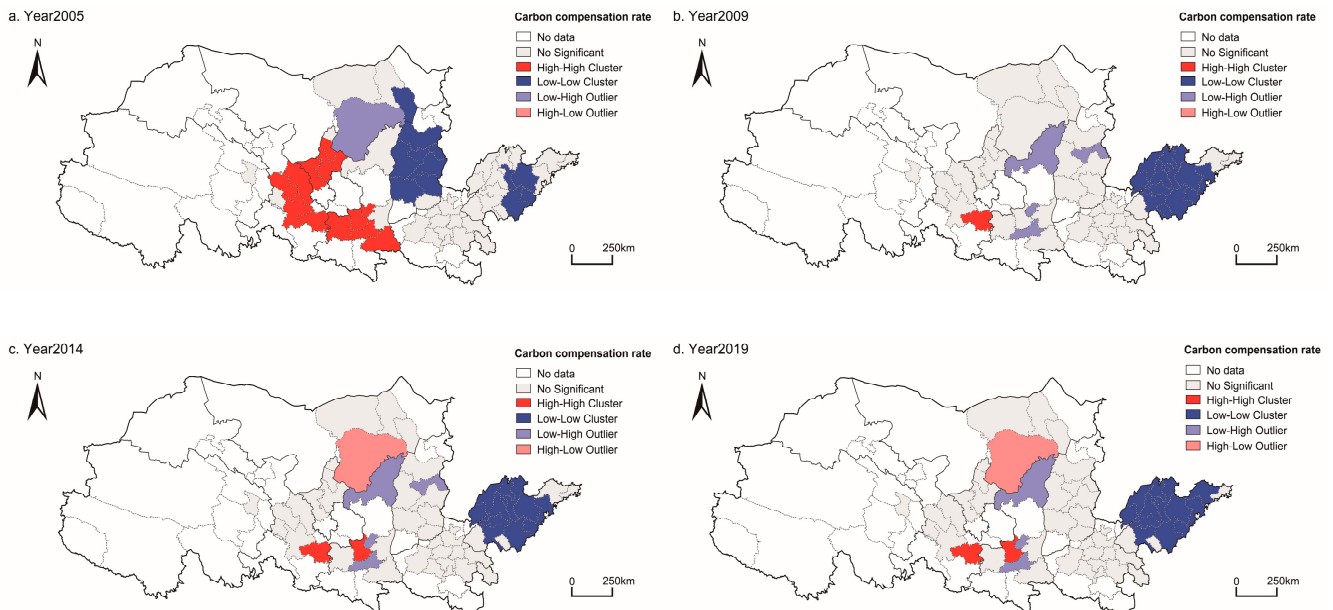

**Figure 6.** LISA agglomeration map of carbon compensation rate in YRB: (**a**) in 2005; (**b**) in 2009; (**c**) in 2014; (**d**) in 2019.

### 4.3.2. β Convergence Test of the Carbon Compensation Rate

The result of the β convergence test of the carbon compensation rate values of the urban agglomerations in the YRB, performed using Stata software, showed that there is a noticeable "catch-up effect". This indicates that the low-value areas of the carbon compensation rate are converging to the high-value areas at a higher growth rate than the high-value areas. Considering the influence of socioeconomic factors on the carbon compensation rate, five variables, namely GDP, population size, urbanization rate, industrial structure, and carbon productivity, were added for the conditional convergence analysis. Tables 3 and 4 show the results, revealing that the carbon compensation rate of urban agglomerations in the YRB converged through absolute and conditional β convergence.

**Table 3.** The results of the absolute convergence tests for the carbon compensation rate of urban agglomerations in the YRB.

| Index | Overall | 1 | 2 | 3 | 4 | 5 | 6 | 7 |
|---|---|---|---|---|---|---|---|---|
| β | −0.1965 *** | −0.0269 *** | −0.1451 *** | −0.4146 *** | −0.0633 *** | −0.1261 *** | −0.0076 *** | −0.1052 *** |
| N | 784 | 70 | 196 | 70 | 56 | 224 | 84 | 84 |
| R2 | 0.0719 | 0.1902 | 0.1411 | 0.4566 | 0.3787 | 0.0395 | 0.0106 | 0.0194 |
| Convergence speed | 0.0156 | 0.0019 | 0.0112 | 0.0382 | 0.0047 | 0.0096 | 0.0005 | 0.0071 |

Note: *** are significant at the 1% level, respectively.

**Table 4.** The results of the conditional convergence tests for the carbon compensation rate of urban agglomerations in the YRB.

| Index | Overall | 1 | 2 | 3 | 4 | 5 | 6 | 7 |
|---|---|---|---|---|---|---|---|---|
| β | −0.4630 *** | −0.5056 *** | −0.2963 *** | −1.0156 *** | −0.9653 *** | −0.5485 *** | −0.2394 *** | −0.3503 *** |
| lnGDP | −0.2777 *** | −0.3962 | −0.1404 | −0.7509 *** | −2.6680 *** | −0.8280 *** | −0.176 | −1.1897 *** |
| lnPOP | 0.3650 ** | 0.4462 | 0.0402 | 0.4093 | 0.0393 | −0.1215 | −0.3183 | 0.9657 * |
| Urbanization rate | −0.0087 *** | −0.0160 * | 0.0034 | 0.0103 | −0.0311* | −0.0126 *** | −0.0320 *** | −0.0142 |
| Industry Structure | 0.0001 | −0.018 | −0.0081 ** | −0.0136 | 0.0374* | 0.0122 *** | −0.0192 *** | −0.0003 |
| Carbon productivity | 0.0798 *** | 0.0924 *** | 0.0348 *** | 0.1574 *** | 0.9125 *** | 0.1920 *** | 0.0816 *** | 0.0486 *** |
| N | 784 | 70 | 196 | 70 | 56 | 224 | 84 | 84 |
| R2 | 0.34 | 0.4833 | 0.4415 | 0.8915 | 0.8641 | 0.1087 | 0.3397 | 0.5515 |
| Convergence speed | 0.0444 | 0.0503 | 0.0251 | 0.0156 | 0.24 | 0.0568 | 0.0195 | 0.0308 |

Note: ***, **, and * are significant at the 1%, 5%, and 10% levels, respectively.

All β coefficients in Table 3 are less than 0, which means that there is convergence. The numbers 1–7 correspond to the Lanxi Urban Agglomeration, Central Plains Urban Agglomeration, Hubao-Eyu Urban Agglomeration, and Ningxia Cities along the Yellow River Group. In terms of the convergence rate, the results of using two analyses were different, showing that these variables considered in the study had an impact on the results. The speed of convergence in the two convergence analyses was highest in the Hubao-Eyu Urban Agglomeration and Ningxia Cities along the Yellow River Group, respectively. In the three time periods of 2005–2009, 2010–2014, and 2015–2019, the convergence rate of the overall carbon compensation rate in the YRB was analyzed, and it was found that the convergence rate gradually slowed down, the specific results are shown in Table 5. Analysis of five conditional variables through conditional convergence results showed that the overall carbon compensation rate in the YRB had a significant negative correlation with GDP and urbanization rate, and had a certain positive correlation with population size and industrial structure, but the results of each urban agglomeration were inconsistent with the overall trend; therefore, the significant relationship between these variables and the carbon compensation rate needs to be demonstrated. However, there was a significant positive relationship between carbon productivity and the carbon compensation rate in the YRB as a whole. The results of each urban agglomeration were consistent with the overall direction. The specific results are shown in Table 6.

**Table 5.** The results of the absolute convergence tests of the carbon compensation rate in the YRB urban agglomeration by stage.

| Index | 2005–2009 | 2010–2014 | 2015–2019 |
|---|---|---|---|
| β | −0.5416 *** | −0.2528 *** | −0.1130 *** |
| N | 280 | 280 | 280 |
| R2 | 0.282 | 0.2826 | 0.2459 |
| Convergence speed | 0.1072 | 0.0772 | 0.0559 |

Note: *** are significant at the 1% level, respectively.

**Table 6.** The results of the conditional convergence tests of the carbon compensation rate in the YRB urban agglomeration by stage.

| Index | 2005–2009 | 2010–2014 | 2015–2019 |
|---|---|---|---|
| β | −0.8712 *** | −0.8737 *** | −0.7739 *** |
| lnGDP | −0.6271 *** | −0.4276 *** | −0.8262 *** |

**Table 6.** *Cont.*

| Index | 2005–2009 | 2010–2014 | 2015–2019 |
|---|---|---|---|
| lnPOP | 0.9316 ** | −0.286 | 0.3174 |
| Urbanization rate | 0.0037 | −0.0176 ** | −0.0252 *** |
| Industry Structure | −0.0115 * | −0.0237 *** | 0 |
| Carbon productivity | 0.1094 *** | 0.1758 *** | 0.0952 *** |
| N | 280 | 280 | 280 |
| R2 | 0.4853 | 0.6699 | 0.7275 |
| Convergence speed | 0.1464 | 0.1478 | 0.1062 |

Note: ***, **, and * are significant at the 1%, 5%, and 10% levels, respectively.

### 4.4. Spatial and Temporal Patterns of Carbon Compensation Potential in Urban Agglomerations in the YRB

#### 4.4.1. Trends in Carbon Compensation Potential over Time

Figure 7 shows the change in carbon compensation potential in the YRB from 2005 to 2019. It can be seen that the carbon compensation potential of most urban clusters was more than 50%, indicating that there is a large potential for carbon compensation to be explored. This is also related to the large difference in carbon compensation rate of urban agglomerations. The overall carbon compensation potential curve of the YRB was steadily rising, indicating that its potential is constantly rising and there is increasingly more room for emission reduction.

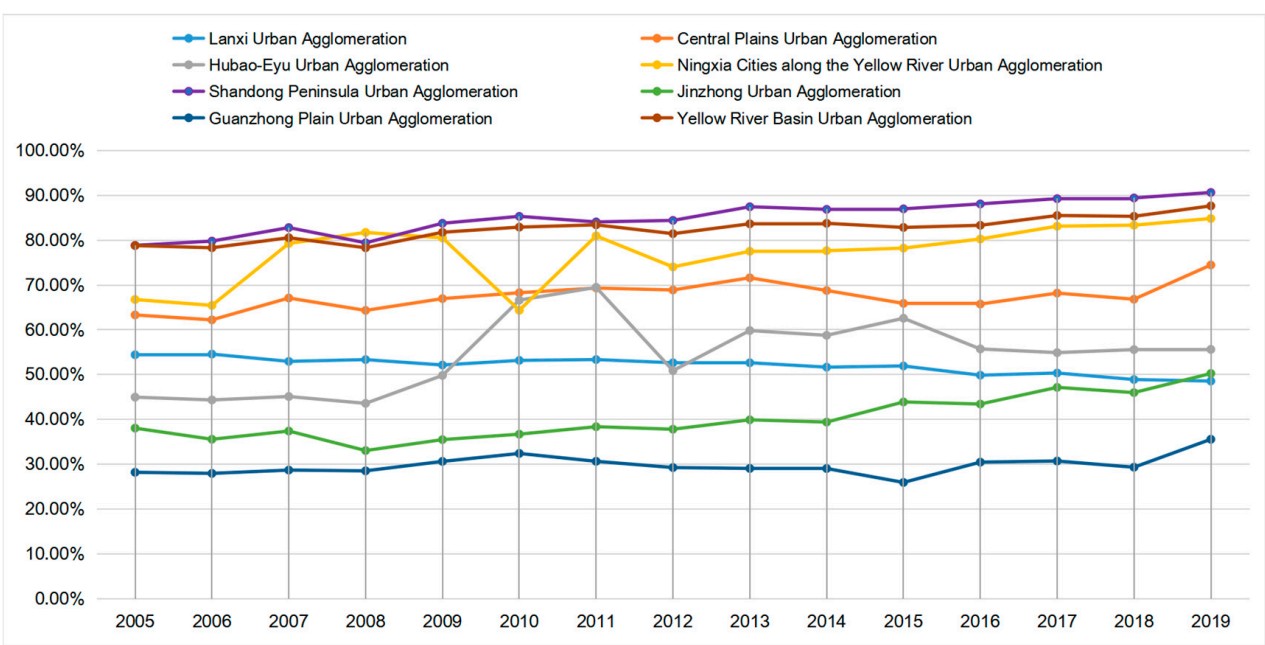

**Figure 7.** Carbon compensation potential of urban agglomerations in the YRB.

#### 4.4.2. Spatial Characteristics of Carbon Compensation Potential

Figure 8 shows the distribution of carbon compensation potential in the YRB in 2005, 2009, 2014, and 2019 using the five-level natural breakpoint method. Overall, the values of the carbon compensation potential of the urban agglomerations in the YRB are increasing with time, and the regions with the most significant changes in spatial patterns are in the eastern part of the study area, mainly in the Shandong Peninsula Urban Agglomeration. The areas with low carbon compensation potential and high–high agglomeration areas are concentrated in the Hubao-Eyu Urban Agglomeration, Lanxi Urban Agglomeration, Guanzhong Plain Urban Agglomeration, and the western and northern areas of the Jinzhong Urban Agglomeration. The higher carbon compensation potential of provincial capitals may be related to the strong socioeconomic ties between these cities and other

cities, and the high population density and urbanization rate within the city. Overall, the eastern region has changed greatly and is a key region for reducing carbon emissions. The carbon compensation potential of the YRB showed a spatial distribution of "low in the west and high in the east", and the change in the western region was small, whereas the eastern region changed greatly. During the sampled period, the growth rate of carbon offset potential in the western region was 15.5%, and the growth rate in the eastern region was 19.6%.

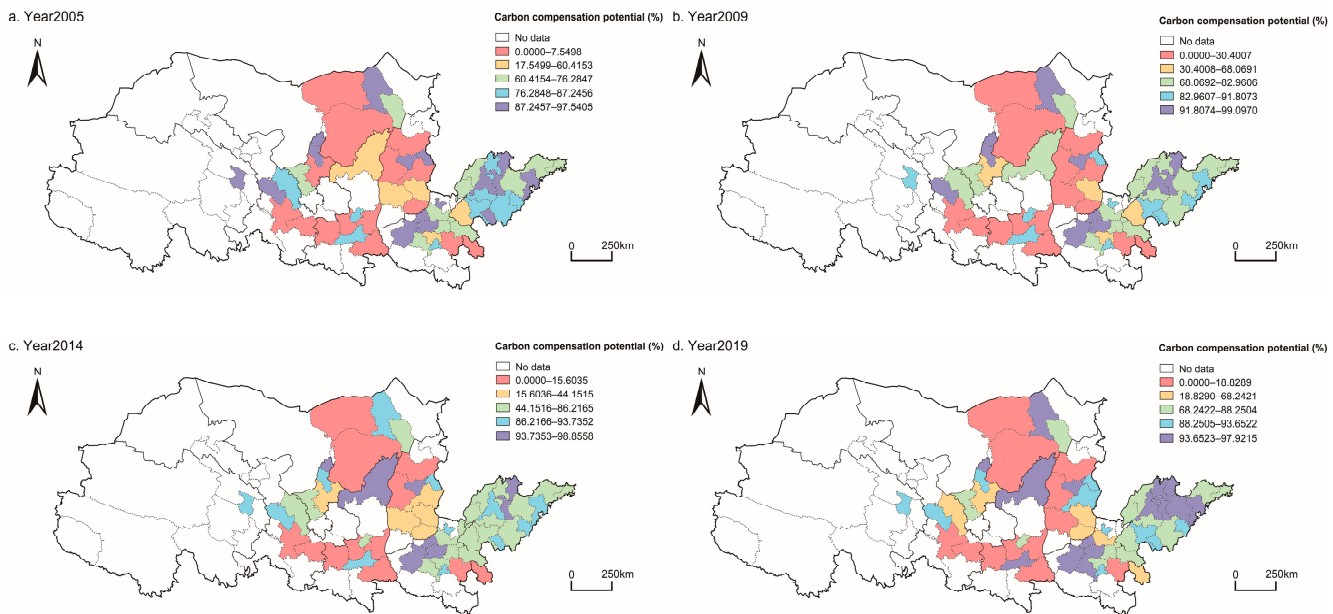

**Figure 8.** Spatial distribution of carbon compensation potential of urban agglomerations in the YRB: (**a**) in 2005; (**b**) in 2009; (**c**) in 2014; (**d**) in 2019.

### 4.5. Network Linkage Study on the Carbon Compensation Potential of Urban Agglomerations in the YRB

4.5.1. Overall Network Analysis of Carbon Compensation Potential

ArcGIS 10.6 software was used to analyze the network correlation of the carbon compensation potential between cities in the YRB in 2005, 2009, 2014, and 2019. The gravity model was modified to make it more suitable for the study of carbon compensation potential correlation between cities, and the five-level natural break point method was used to visualize the gravitational lines hierarchically for the study. The weak gravitational value correlation lines between cities were deleted, and the obtained results are shown in Figure 9. From the figure, it is evident that the number of gravitational lines increased, as did the number of strongly associated gravitational lines, indicating that the network association of carbon compensation potential between cities is strengthening. The figure indicates that the gravitational association of carbon compensation potential among cities in the YRB mainly occurred in the Shandong Peninsula Urban Agglomeration and the Central Plains Urban Agglomeration. Through the preliminary characterization of the network structure of carbon compensation potential in the YRB, we understand the basic characteristics of the gradual strengthening of the overall network correlation and the strong correlation between the urban agglomeration of the Shandong Peninsula and the Central Plains Urban Agglomeration. Therefore, we have further analyzed these strongly correlated urban agglomerations to explore the internal mechanism of their network association.

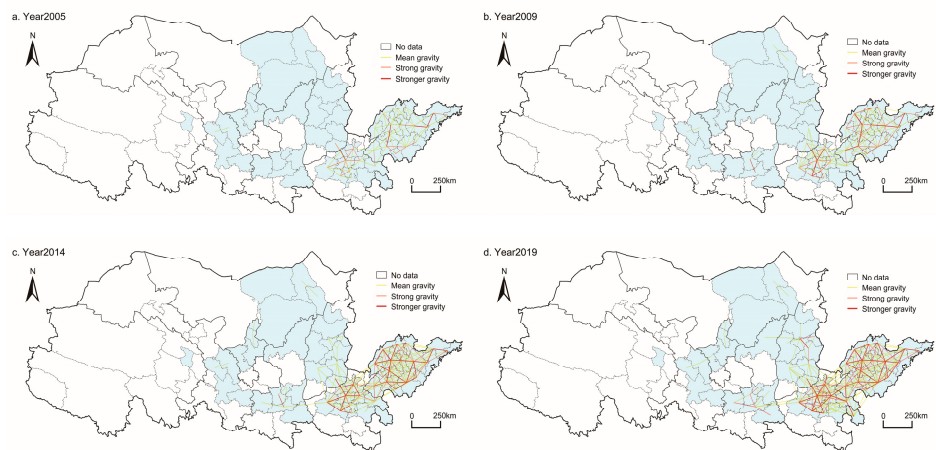

**Figure 9.** Overall network linkage of carbon compensation potential of urban agglomerations in the YRB: (**a**) in 2005; (**b**) in 2009; (**c**) in 2014; (**d**) in 2019.

4.5.2. Cohesive Subgroup Analysis

The carbon compensation potential of the Shandong Peninsula, the Central Plains, and the Guanzhong Urban Agglomeration have a high spatial correlation, so they are further analyzed as cohesive subgroups. Figure 10 shows the results of the chunk-type analysis of the annual 2019 carbon compensation potential values for each of the three urban agglomerations. Using the CONCOR tool, the first two are divided into four clusters, and the latter are divided into three clusters.

(1)    Urban agglomeration network characteristics

Figure 10a shows the results of network density, network connection, network hierarchy, and network efficiency for the three urban agglomerations analyzed for the YRB and the cohesive subgroups. First, looking at the overall YRB, the network compactness increased slightly during 2005–2019, but the network level was low, and the network structure was not sufficiently robust. Then, the Shandong Peninsula Urban Agglomeration was analyzed, and its network structure had no isolated points; however, there was no significant relationship hierarchy, indicating that the radiation effect of node central cities on surrounding cities was not obvious. Next, the Central Plains Urban Agglomeration was analyzed, and its network correlation was tighter, the number of isolated points was reduced, the redundant lines were reduced, and the network structure was stable. Finally, the network density and correlation of the urban agglomeration of the Guanzhong Plain decreased; however, the radiation driving effect of the central city on the surrounding cities was better than that of other urban agglomerations.

(2)    Subgroup attribute characteristics

Table 7 shows the correlation indicators of the three city cluster chunking types, and the density matrix and image matrix generated for each cluster. Compared with the overall network density, if the value in the density matrix is higher than the average density, the value is 1 and the inverse value is 0, resulting in the image matrix [39]. The number of plate relationships and the overall trend were analyzed. If the number of intra-board relationships is greater than the number of inter-plate relationships, it indicates that the intra-board relationship is stronger than the inter-plate relationship. The analysis shows that the internal connection between the urban agglomeration of Shandong Peninsula and the urban agglomeration of the Central Plains is stronger, while the subgroups of the Guanzhong Urban Agglomeration are more interconnected.

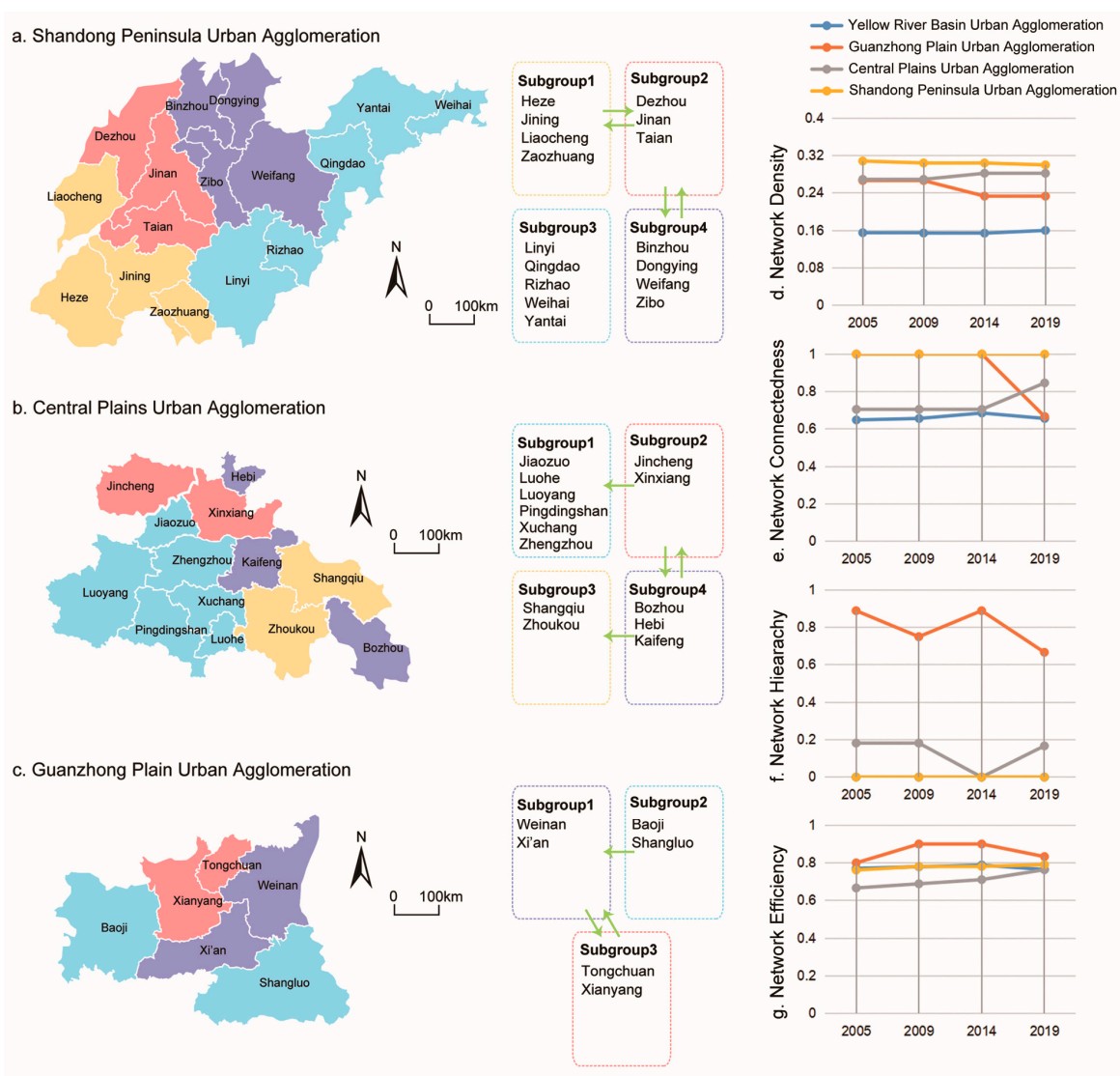

**Figure 10.** Block-type results for three urban agglomerations and related index line graphs: (**a**) Shangdong Peninsula Urban Agglomeration; (**b**) Central Plains Urban Agglomeration; (**c**) Guanzhong Urban Agglomeration; (**d**) Network Density; (**e**) Network Connection; (**f**) network hierarchy; (**g**) network efficiency.

**Table 7.** Relevant indicators for the three urban agglomerations by block type.

| Subgroup | Reception Relation Matrix | | | | Number of Receiving Relations | Number of Overflow Relations | Prpportion of Expected Internal Relationship (%) | Prpportion of Actual Internal Relationship (%) | Density Matrix | | | | Image Matrix | | | |
|---|---|---|---|---|---|---|---|---|---|---|---|---|---|---|---|---|
| | 1 | 2 | 3 | 4 | | | | | 1 | 2 | 3 | 4 | 1 | 2 | 3 | 4 |
| Shandong Peninsula Urban Agglomeration | | | | | | | | | | | | | | | | |
| 1 | 8 | 5 | 2 | 0 | 7 | 7 | 46.67 | 53.33 | 0.67 | 0.42 | 0.10 | 0.00 | 1 | 1 | 0 | 0 |
| 2 | 5 | 4 | 3 | 5 | 13 | 13 | 20.00 | 23.53 | 0.42 | 0.67 | 0.20 | 0.42 | 1 | 1 | 0 | 1 |
| 3 | 2 | 3 | 11 | 4 | 10 | 9 | 66.67 | 55.00 | 0.10 | 0.20 | 0.55 | 0.20 | 0 | 0 | 1 | 0 |
| 4 | 0 | 5 | 5 | 10 | 9 | 10 | 60.00 | 50.00 | 0.00 | 0.42 | 0.25 | 0.83 | 0 | 1 | 0 | 1 |
| Central Plains Urban Agglomeration | | | | | | | | | | | | | | | | |
| 1 | 24 | 5 | 1 | 2 | 5 | 8 | 191.67 | 75.00 | 0.80 | 0.42 | 0.08 | 0.11 | 1 | 1 | 0 | 0 |
| 2 | 3 | 0 | 0 | 2 | 7 | 5 | −8.33 | 0.00 | 0.25 | 0.00 | 0.00 | 0.33 | 0 | 0 | 0 | 1 |
| 3 | 0 | 0 | 0 | 2 | 2 | 2 | −8.33 | 0.00 | 0.00 | 0.00 | 0.00 | 0.33 | 0 | 0 | 0 | 1 |
| 4 | 2 | 2 | 1 | 0 | 6 | 5 | −8.33 | 0.00 | 0.11 | 0.33 | 0.17 | 0.00 | 0 | 1 | 0 | 0 |
| Guanzhong Plain Urban Agglomeration | | | | | | | | | | | | | | | | |
| 1 | 0 | 2 | 2 | | 2 | 4 | 20.00 | 0.00 | 0.00 | 0.50 | 0.50 | | 0 | 1 | 1 | |
| 2 | 0 | 0 | 0 | | 0 | 0 | −20.00 | 0.00 | 0.00 | 0.00 | 0.00 | | 0 | 0 | 0 | |
| 3 | 2 | 0 | 1 | | 2 | 2 | 20.00 | 83.33 | 0.50 | 0.00 | 0.50 | | 1 | 0 | 1 | |

The number of receiving and overflow relations in Table 7 can represent the intensity of feature inflow and outflow within a subgroup [13]. Subgroups 1 and 2 of the urban agglomeration of Shandong Peninsula have the same number of receiving and sending relationships, and the actual internal relationship is larger than expected, so they belong to the two-way spillover plate; subgroup 3 has a more significant proportion of inflow intensity and expectation relationship, which belongs to the net income sector; subgroup 4 has more relationships and a large proportion of expected relationships, so it belongs to the net spillover plate. The outflow intensity and expectation of subgroup 1 of the Central Plains Urban Agglomeration have a large proportion of outflow intensity and expectation, which belongs to the net spillover plate; subgroups 2 and 4 have a large proportion of inflow intensity and actual expectations, so they belong to the net income sector; subgroup 3 receives and emits the same number of relations and belongs to the broker sector. Subgroup 1 of the urban agglomeration in the Guanzhong Plain has a large ratio of outflow intensity and expectation, which belongs to the net spillover plate; subgroups 2 and 3 have the same number of receiving relations and issuing relations and belong to the broker sector.

(3)    Inter-subgroup correlation effects

Combined with other indicators of each plate of the three urban agglomerations, the spillover and income effects were analyzed. If the actual relationship ratio is larger than expected, there may be a spillover effect, and if it is less, there may be a benefit effect. This needs to be determined in combination with the image matrix value. Firstly, the urban agglomeration of the Shandong Peninsula was analyzed, and it was found that there are spillover effects from subgroup 1 to subgroup 2, from subgroup 2 to subgroup 1 and subgroup 4, and from subgroup 4 to subgroup 2, and there is no transmission path between subgroup 3 and other clusters. From this point of view, the second group is the core of the urban agglomeration, which includes the three cities of Dezhou, Jinan, and Tai'an, and has greater carbon compensation potential improvement; subgroup 3 is relatively dispersed geographically, and the connection was weak in the overall network association, and there was no obvious spillover and benefit effect. Next, the Central Plains Urban Agglomeration was analyzed, and it was found that subgroup 2 received the spillover effect of subgroup 1 and subgroup 4, and subgroup 4 received the spillover effect of subgroup 2 and subgroup 3, and the internal relationship of subgroup 2, 3, and 4 was not obvious. The two clusters spanning Shanxi, Henan, and Anhui provinces were geographically discontinuous, but had a two-way spillover relationship, indicating that the network correlation of carbon compensation potential breaks through administrative boundaries and reaffirms the need to study regional issues through urban agglomerations. Each group had conduction paths connected to each other, with a more rigorous network structure than the urban agglomeration of the Shandong Peninsula. Finally, the urban agglomeration of the Guanzhong Plain was analyzed, and a spillover effect was found for subgroup 1 receiving subgroup 3, subgroup 2 receiving subgroup 1, and subgroup 3 receiving subgroup 1. Subgroups 1 and 3 had bidirectional transmission paths, which are also geographically concentrated, and are also the areas through which the Wei River, the largest tributary of the Yellow River, flows. This is an area that should be of concern to policymakers.

### 4.5.3. Individual Centrality Analysis

The degree of centrality, closeness centrality, and between-ness centrality of each city in 2005, 2009, 2014, and 2019 were analyzed. For the sake of space, Table 8 only shows the top ten cities in the analysis results for each indicator and the respective city clusters in which they are located.

**Table 8.** Results of the analysis of centrality indicators for each city.

| Year | 2005 | | | 2009 | | | 2014 | | | 2019 | | |
|---|---|---|---|---|---|---|---|---|---|---|---|---|
| **Index** | **City Cluster** | **Number** | **City** | **City Cluster** | **Number** | **City** | **City Cluster** | **Number** | **City** | **City Cluster** | **Number** | **City** |
| **Point-In Degree Centrality** | Central Plains Urban Agglomeration | 3 | Zhengzhou, Shangqiu, Xinxiang | Central Plains Urban Agglomeration | 3 | Zhengzhou, Shangqiu, Xinxiang | Central Plains Urban Agglomeration | 3 | Zhengzhou, Shangqiu, Xinxiang | Central Plains Urban Agglomeration | 3 | Zhengzhou, Shangqiu, Xinxiang |
| | Shandong Peninsula Urban Agglomeration | 7 | Jining, Liaocheng, Heze, Jinan, Linyi, Taian, Dezhou | Shandong Peninsula Urban Agglomeration | 7 | Jining, Liaocheng, Heze, Jinan, Linyi, Taian, Dezhou | Shandong Peninsula Urban Agglomeration | 7 | Jining, Liaocheng, Heze, Jinan, Linyi, Taian, Weifang | Shandong Peninsula Urban Agglomeration | 7 | Jining, Liaocheng, Heze, Jinan, Linyi, Taian, Weifang |
| **Point-Out Degree Centrality** | Central Plains Urban Agglomeration | 4 | Zhengzhou, Xinxiang, Shangqiu, Kaifeng | Central Plains Urban Agglomeration | 4 | Zhengzhou, Xinxiang, Shangqiu, Kaifeng | Central Plains Urban Agglomeration | 4 | Zhengzhou, Xinxiang, Shangqiu, Kaifeng | Central Plains Urban Agglomeration | 4 | Zhengzhou, Jiaozuo, Shangqiu, Kaifeng |
| | Shandong Peninsula Urban Agglomeration | 6 | Jining, Jinan, Liaocheng, Linyi, Taian, ZaoZhuang | Shandong Peninsula Urban Agglomeration | 6 | Jining, Jinan, Liaocheng, Linyi, Taian, Heze | Shandong Peninsula Urban Agglomeration | 6 | Jining, Jinan, Liaocheng, Linyi, Taian, Heze | Shandong Peninsula Urban Agglomeration | 6 | Jining, Jinan, Liaocheng, Linyi, Taian, Heze |
| **Point-In Closeness Centrality** | Central Plains Urban Agglomeration | 3 | Zhoukou, Bozhou, Zhengzhou | Central Plains Urban Agglomeration | 4 | Zhoukou, Bozhou, Zhengzhou, Xinxiang | Central Plains Urban Agglomeration | 4 | Zhoukou, Bozhou, Zhengzhou, Xinxiang | Central Plains Urban Agglomeration | 2 | Zhoukou, Zhengzhou |
| | | | | | | | | | | Guanzhong Plain Urban Agglomeration | 1 | Shangluo |
| | Hubao-Eyu Urban Agglomeration | 2 | Hohhot, Baotou | Hubao-Eyu Urban Agglomeration | 2 | Hohhot, Baotou | | | | Hubao-Eyu Urban Agglomeration | 2 | Hohhot, Baotou |
| | Jinzhong Urban Agglomeration | 2 | Jinzhong, Xinzhou | Jinzhong Urban Agglomeration | 1 | Jinzhong | Jinzhong Urban Agglomeration | 3 | Xinxiang, Linfen, Lvliang, Xinzhou | Jinzhong Urban Agglomeration | 3 | Linfen, Lvliang, Xinzhou |
| | Shandong Peninsula Urban Agglomeration | 3 | Jining, Liaocheng, Heze | Shandong Peninsula Urban Agglomeration | 3 | Jining, Liaocheng, Heze | Shandong Peninsula Urban Agglomeration | 3 | Jining, Liaocheng, Heze | Shandong Peninsula Urban Agglomeration | 2 | Heze, Jinan |

**Table 8.** *Cont.*

| Year | 2005 | | | 2009 | | | 2014 | | | 2019 | | |
|---|---|---|---|---|---|---|---|---|---|---|---|---|
| **Index** | City Cluster | Number | City | City Cluster | Number | City | City Cluster | Number | City | City Cluster | Number | City |
| **Point-Out Closeness Centrality** | Central Plains Urban Agglomeration | 6 | Zhengzhou Jiaozuo Xinxiang Luoyang Shangqiu Kaifeng | Central Plains Urban Agglomeration | 5 | Zhengzhou Xinxiang Luoyang Shangqiu Kaifeng | Central Plains Urban Agglomeration | 4 | Zhengzhou Xinxiang Luoyang Shangqiu | Central Plains Urban Agglomeration | 5 | Zhengzhou Xinxiang Luoyang Shangqiu Jiaozuo |
| | Jinzhong Urban Agglomeration | 1 | Taiyuan | | | Jining | | | Jining Jinan | | | Jining |
| | Shandong Peninsula Urban Agglomeration | 3 | Jining Liaocheng Linyi | Shandong Peninsula Urban Agglomeration | 5 | Jinan Liaocheng Heze Linyi | Shandong Peninsula Urban Agglomeration | 6 | Liaocheng Heze Linyi Taian | Shandong Peninsula Urban Agglomeration | 5 | Jinan Liaocheng Linyi Taian |
| **Betweenness centrality** | Central Plains Urban Agglomeration | 3 | Zhengzhou Jiaozuo Shangqiu | Central Plains Urban Agglomeration | 3 | Zhengzhou Luoyang Shangqiu | Central Plains Urban Agglomeration | 3 | Zhengzhou Xinxiang Shangqiu | Central Plains Urban Agglomeration | 4 | Zhengzhou Luoyang Shangqiu Jiaozuo |
| | Guanzhong Plain Urban Agglomeration | 1 | Xi'an | Guanzhong Plain Urban Agglomeration | 1 | Xi'an | Guanzhong Plain Urban Agglomeration | 1 | Xi'an | Guanzhong Plain Urban Agglomeration | 1 | Xi'an |
| | Jinzhong Urban Agglomeration | 2 | Taiyuan Changzhi | Jinzhong Urban Agglomeration | 2 | Taiyuan Changzhi | Hubao-Eyu Urban Agglomeration | 1 | Hohhot | Jinzhong Urban Agglomeration | 1 | Taiyuan |
| | | | | | | | Jinzhong Urban Agglomeration | 2 | Taiyuan Changzhi | | | |
| | Shandong Peninsula Urban Agglomeration | 4 | Jining Jinan Liaocheng Linyi | Shandong Peninsula Urban Agglomeration | 4 | Jining Jinan Liaocheng Heze | Shandong Peninsula Urban Agglomeration | 3 | Jining Liaocheng Heze | Shandong Peninsula Urban Agglomeration | 4 | Jinan Heze Jining Weifang |

First, we analyzed the degree of centrality, which includes point-in and point-out degrees of centrality. Point-in centrality can measure the attractiveness of cities, and point-out centrality represents the strength of ties that exist with other cities. The top ten cities in the sampled period of point-in centrality and point-out centrality were distributed in the Central Plains Urban Agglomeration and Shandong Peninsula Urban Agglomeration. Of these, the Central Plains Urban Agglomeration contains three cities, and the Shandong Peninsula Urban Agglomeration contains seven cities. However, these seven cities changed slightly, after 2014, when Dezhou withdrew from the top ten ranks, and the new addition was Weifang City, indicating that the cities with strong connection are developing at a rate greater than or equal to other cities. Furthermore, the attractiveness between cities in the eastern part of the Shandong Peninsula Urban Agglomeration is rapidly rising. In general, the Central Plains Urban Agglomeration and Shandong Peninsula Urban Agglomeration have higher centrality. These two urban agglomerations have strong connections and attractions with other cities.

Secondly, the closeness centrality of the city was analyzed. Closeness centrality is similar to degree centrality in that both can be used to measure how closely a city is connected to other individuals and centrality in the network, except that the former is expressed as the average distance between individuals, and the latter is expressed in terms of the number of relationships. In contrast, the latter is expressed in terms of the number of relationships. Closeness centrality can indicate the speed of generating ties, while degree centrality can indicate the strength of ties. A high inward centrality indicates that one's behavior can quickly change the behavior of other neighboring individuals and is central to the network. A high degree of extroverted proximity centrality demonstrates active feedback to changes in neighboring individuals, thus placing such cities in a subordinate position in the network [25]. The top ten cities in terms of inward centrality are located in the Central Plains Urban Agglomeration, Jinzhong Urban Agglomeration, Shandong Peninsula Urban Agglomeration, Hubao-Eyu Urban Agglomeration, Jinzhong Urban Agglomeration, and Guanzhong Plain Urban Agglomeration, which have a small number of cities that satisfy the condition. The distribution pattern of these cities during the sample period shows that they are firstly scattered, then gradually clustered in Henan, and finally scattered in urban agglomerations. This indicates there are scattered core cities in the study area that have highly efficient linkage effects on the neighboring cities and contribute to the close connection and development of the network. In the study of carbon reduction planning, these cities can be used as critical nodes for rational planning. The top ten cities in terms of outward proximity to the center were mainly located in the Central Plains Urban Agglomeration and Shandong Peninsula Urban Agglomeration, which includes Zhengzhou, Xinxiang, and Shangqiu in the Central Plains Urban Agglomeration. The three cities in the Central Plains Urban Agglomeration are also cities with a high degree of centrality, which indicates that these cities not only have more connections but also shorter average distances than other cities, and thus have higher network connection efficiency.

Finally, the intermediary centrality was analyzed, which was used to indicate the number of shortest-distance connections and to measure the efficiency and strength of the connections with other nodes. The top ten cities in intermediary centrality were mainly located in the Central Plains Urban Agglomeration, Guanzhong Plain Urban Agglomeration, Jinzhong Urban Agglomeration, and Shandong Peninsula Urban Agglomeration; the nodes were more scattered in the middle and lower reaches of the Yellow River, indicating that the efficiency and strength of network connections are greater in these regions. In addition, the top ten cities in the intermediary centrality gradually clustered toward the Central Plains Urban Agglomeration and Shandong Peninsula Urban Agglomeration over the course of the sampled period.

In general, several cities with higher centrality indices, such as Zhengzhou, Shangqiu, and Liaocheng, are mainly distributed in the Central Plains Urban Agglomeration and Shandong Peninsula Urban Agglomeration, indicating the existence of more network nodes and centers of gravity in these urban agglomerations. Among the top ten statistics of all

centrality indicators, 48.5% are from the Shandong Peninsula urban agglomeration and 37% are from the Central Plains urban agglomeration. In contrast, Ningxia Cities along the Yellow River Urban Agglomeration had no cities ranked in the top ten in each centrality index, which indicates that this region contributes less to the network and is not closely connected with other urban agglomerations.

## 5. Discussion

After the 2015 Paris climate conference, more and more scholars began to pay attention to the issue of carbon neutrality, and the research on the potential of carbon compensation has gradually been enriched. He et al. found that China's carbon compensation potential showed a spatial pattern of "high in the northwest and low in the southeast". [20]. The results of the spatial layout of the carbon compensation potential studied by us are the same as those of He et al. On this basis, we made improvements not only to analyzing the spatial pattern of carbon compensation potential in YRB but also added SNA methods to study the spatial correlation between urban agglomerations, spillover and benefit effects within urban agglomerations, and the centrality of urban individuals. Through SNA, we can understand on a deeper level whether the radiation driving effect between cities is sufficient, and what the commonalities of cities with higher centrality are. In this study, the carbon compensation potential was calculated using a β convergence test and the parameter comparison method. In addition, the ability of an urban carbon sink surplus to compensate neighboring areas was quantified, and the SNA method was used to study the spatiotemporal pattern and network structure of the carbon compensation potential of urban agglomerations in the YRB, promoting the research of carbon compensation and broaden the application scope of the social network analysis method, which has certain practical significance. However, the data in this study are accurate to the city, and if they are accurate to the county, it should lead to more correct results.

The carbon emissions of the YRB are increasing year by year, which may be related to the large proportion of resource-based cities and old industrial cities in the YRB [42]. The carbon compensation rate and carbon compensation potential are "upstream high and downstream low" and "upstream low and downstream high", respectively. Furthermore, the urban agglomerations in the YRB have different resource endowment conditions, and their economic condition and ecological environment status are quite different. Moreover, the development level is distributed in a stepwise manner, and the advantages are not the same [43]. At present, it is quite difficult for the western region to break through the existing basic conditions and achieve all-round development, and there is still a gap between the development of the western region and the development of the upstream region [43,44]. Moreover, what we currently agree upon is that the development process of cities is fast, and the high quality of development means increased carbon emissions, which may be correlated to the dense population of large cities [40]. Furthermore, low-carbon technology should be innovative, and the current energy structure is not perfect. The study finds that provincial capitals generally have high carbon compensation potential, indicating that these cities have greater progress space and the potential of radiation-driven effect in surrounding cities [45]. China's industrialization and urbanization have accelerated, and economic growth has brought benefits such as improved infrastructure and enhanced external links with cities [46], making urban agglomerations more connected.

Based on the research results, three suggestions were made for the construction of YRB: (1) To strengthen the connection between cities in the upper reaches of the YRB. In terms of the middle and lower reaches of the YRB, we should continue to build projects for the export of energy from the west to the east, such as "west-to-east electricity transmission" and "west-to-east gas transfer". We should also sufficiently play to the strengths of upstream natural resource endowments and promote an overall virtuous cycle and high-quality development of the YRB. (2) The carbon compensation rate of cities in the YRB shows a certain agglomeration trend. Therefore, regional governance models can be considered, such as those based on urban agglomerations or the agglomeration mode,

and the governance models of each region can be proposed separately to achieve accurate problem-solving. (3) Taking advantage of the current network structure, the city with strong centrality in the research area is regarded as the node city of regional governance, and the radiation pulling role of the node city on the surrounding city should allow the formation of a hierarchical and efficient governance system.

## 6. Conclusions

In this study, the spatiotemporal pattern of the carbon compensation rate and carbon compensation potential in the YRB was analyzed with the help of GIS, GeoDa, Stata, and other spatiotemporal research tools. In addition, the kernel density curve was used to fit its evolution characteristics. The carbon compensation potential was calculated using the $\beta$ convergence test and parameter comparison method, and the social network analysis method was used to study the cyberspatial association of carbon compensation potential in the YRB. The conclusions of the study are as follows:

(1) The carbon compensation rate in the YRB shows a pattern of "high in the northwest and low in the southeast". Furthermore, the carbon compensation rate presents a downward trend during 2005–2019. The spatial distribution of carbon offset rates in YRB has roughly expanded along the east-west axis, and in 2019, compared with 2005, the distribution axis shifted $0.21°$ to the northwest, and the center shifted by about 339km. Based on fitting using the kernel density curve, it is found that the development of the YRB carbon compensation rate shows a trend of weak divergence and polarization.

(2) The carbon compensation rate between cities had a spatial correlation and a "catch-up effect". Moreover, it is found that the YRB carbon compensation rate is significantly negatively associated with GDP and urbanization rate and markedly positively correlated with carbon productivity, which is significant at the 1% level.

(3) Overall, the carbon compensation potential of the YRB is obviously higher in the east than in the west. In 2019, the average carbon compensation potential of the eastern part of the YRB was 15.6% higher than that of the western region. Moreover, the carbon compensation potential increased over time, with little change in the west and a bigger change in the east. The carbon compensation potential of the urban agglomeration of the Shandong Peninsula is growing faster than other regions in the YRB.

(4) The network correlation intensity of carbon compensation potential in the YRB increased significantly during 2005–2019, and the correlation in the downstream region was stronger than that in the upstream region. The correlation within the subgroups of the Shandong Peninsula Urban Agglomeration and the Central Plains Urban Agglomeration was stronger than that between subgroups, and the inter-group correlation of the Guanzhong Urban Agglomeration was much greater than that within the subgroup. The network structure of the Central Plains Urban Agglomeration is the most rigorous.

(5) The analysis of urban centrality shows that there are many network centers in the Shandong Peninsula and the Central Plains urban agglomeration, including Liaocheng, Zhengzhou, Shangqiu, and other cities. Among the top ten statistics of all centrality indicators, 48.5% are from the Shandong Peninsula Urban Agglomeration and 37% are from the Central Plains Urban Agglomeration. The centrality index of cities along the Yellow River in Ningxia is low, and other urban agglomerations are not closely connected.

(6) The YRB should continue to give full play to its upstream resource advantages and carry out projects such as "west-to-east electricity transmission" and "west-to-east gas transfer" to reduce downstream energy pressure. Areas with low carbon compensation rates and cities with high carbon offset potential should accelerate the low-carbon transition. The spatial model of central cities should be used to drive overall development efficiency and coordination to effectively manage carbon reduction tasks.

**Author Contributions:** Project administration, Supervision, Haihong Song; Methodology, Software, Visualization, Writing—original draft, Writing—review and editing, Yifan Li; Data curation, Liyuan Gu; Resources, Jingnan Tang; Validation, Xin Zhang. All authors have read and agreed to the published version of the manuscript.

**Funding:** This research is supported by the Opening Fund of Key Laboratory of Interactive Media Design and Equipment Service Innovation, Ministry of Culture and Tourism (2020 + 11).

**Institutional Review Board Statement:** Not applicable.

**Informed Consent Statement:** Not applicable.

**Data Availability Statement:** The data involved in this study are available from the corresponding authors.

**Conflicts of Interest:** The authors declare no conflict of interest.

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
