# Peer review of "Spatiotemporal Pattern of Carbon Compensation Potential and Network Association in Urban Agglomerations in the Yellow River Basin"

_ijgi, doi:10.3390/ijgi12100435_

Round 1

Reviewer 1 Report

Dear Authors,

The work is interesting and new. I suggest you some modifications as follow, and English revision.

Abstract: please report the results in numbers.

Introduction: specify all the acronyms through the manuscript, including YRB, DMSP-OLS NTL and NPP-VIIRS NTL. When you write about YRB add a reference right after you state that it emits 1.6 times more carbon than the country. Specify to what researches you refer about carbon emissions, sequestration, compensation and compensation potential.

Methods: Correct COR>1: I think it should be CCR.

3.2 section (3): is it COR max or CCR max? it is unclear.

Results: Specify the meaning of asterisks in the Tables.

The authors Song et al. deals with an interesting topic related to carbon compensation potential and network associations in urban Chinese agglomerations located in the Yellow River Basin. The research focuses on the spatio-temporal pattern of carbon compensation potential. The manuscript does not have line numbering as it is not easy to review. I think that the manuscript is new and is publishable. It only needs some minor revisions, to be formatted and edited for English language. 

Hence, I suggest minor revision before considered for publication.

Regards

Dear Authors,

The work is interesting and new. I suggest you some modifications as follow, and English revision.

Abstract: please report the results in numbers.

Introduction: specify all the acronyms through the manuscript, including YRB, DMSP-OLS NTL and NPP-VIIRS NTL. When you write about YRB add a reference right after you state that it emits 1.6 times more carbon than the country. Specify to what researches you refer about carbon emissions, sequestration, compensation and compensation potential.

Methods: Correct COR>1: I think it should be CCR.

3.2 section (3): is it COR max or CCR max? it is unclear.

Results: Specify the meaning of asterisks in the Tables.

1.What is the main question addressed by the research?

Carbon compensation and carbon compensation potential in the urban agglomerations of the Yellow River Basin.

2. Do you consider the topic original or relevant in the field? Does it

address a specific gap in the field?

I think it is relevant in the field as it provides results and specific information related that area.

3. What does it add to the subject area compared with other published

material?

The easy way of application.

4. What specific improvements should the authors consider regarding the

methodology? What further controls should be considered?

See my comments.

5. Are the conclusions consistent with the evidence and arguments presented

and do they address the main question posed?

Yes

6. Are the references appropriate?

They should be improved.

7. Please include any additional comments on the tables and figures.

The meaning of the asterisks should be explained. In addition, the parameters in extenso should be reported in the captions.

The work needs minor editing of English language.

Reviewer 2 Report

This manuscript requires major revision:

Q1:

The innovations of this work should be emphasized in the abstract.

Q2:

It would be more readable to move the carbon sequestration calculation steps to the Methodology section 

.

Q3:

Can the authors explain the basis or reason for choosing these 5 variables in equation 3?

Q4:

I’m confused about the conception of Fij. What is the actual meaning corresponding to the Fij. It would be better to explain it in the text.

Q5:

What is the difference between “urban clusters” and “urban agglomerations”?

Q6:

“group” and “relationship” suddenly appeared in the Section (2) Cohesive subgroup analysis. These conceptions should be defined or described. We want to know how is this gravitational network analyzed in UCINET6, e.g., what do “intra-plate links” refer to?

Q7:

The results section would be better served by adjusting the length of the sub-sections to highlight the key results of the paper.

The grammar of scientific writing needs to be improved. For example, “double carbon goal” is a Chinese oral expression that should be used more accurately to be understood by more scholars internationally. 

This sentence “the spatiotemporal situation and network correlation research potential of the urban agglomeration ...... were spatiotemporal” seems redundant. Is the conjunction "and" used to indicate the parallelism of "potential" and "spatial"?

It is better to express the same semantics with the same words. For example, “carbon offset potential” and “carbon compensation potential” both appear in this manuscript.

It is recommended to enhance the English writing of the manuscript.

Reviewer 3 Report

In General:

Thank you very much for addressing a vital issue of the current world, specifically the global south region, China. This study use β convergence tests and parameter comparison methods to assess the carbon compensation capacity of cities in the Yellow River Basin, China. This study examines the features of network associations and their importance among metropolitan agglomerations. The findings indicate a consistent decline, a notable geographical relationship, and an intensified network association, with the Shandong Peninsula exhibiting the most prominent network hub.

However, this paper can be revised to meet the standard. Specifically, language needs to be improved and coherent, and references must be complete. The introduction and conclusion sections need proper revision to match the target of the study and results. This paper is not yet up to the mark to be published. The manuscript should be edited by a professional editor as there are many mistakes in grammatical range and accuracy. See my specific comments below:

Title

It could be “Spatiotemporal Variability of Carbon Compensation Potential and Network Interactions in Urban Agglomerations of the Yellow River Basin, China”. Please change it if appropriate.

Abstract

·         There is a lack of context and motivation behind the research (in the abstract). 

·         Please do not use any short form in the abstract

·         The implications of the study in the abstract are not spelled out.

·         Follow the IMRAD format for writing the abstract.

Introduction

·         Is there any evidence for your problem statement? In order to ensure clarity and avoid misleading the reader, it is necessary to provide a comprehensive explanation or elaboration when discussing certain circumstances.

·         There is a concept gap for the “dual carbon goal”. Never expect your audience to know everything.

·         There is a lack of context and motivation behind this study. Research Gaps are not well identified. I mean, why are you going to do this research? You may argue that carbon compensation is often used in regions of the global north, but it presents a distinctive challenge when applied to China. Like this….

·         You must compare your methods with others to justify why you chose them.

·         Please look at your objectives and see your conclusion; I think you will not find all the answers. The coordination of objectives and conclusions is essential in scientific papers.

Materials and Methods

·         How can you classify the data? What is the acceptability ratio of image classification?

·         How did you fix the analysis year? Unless there is a clear justification, utilizing data from the year 2019 for analysis or research purposes in the year 2023 is not advisable.

·         What are the limitations of your methods? How can you overcome this?

Results

·         Without proper validation, it is very tough to rely on your results.

Discussion and Conclusion:

·         In the discussion, you need to write a few comparisons between your findings and others.

·         The limitations of your work and future research

Moderate editing of English language required

Reviewer 4 Report

Dear Author(s),

I hope you are well.

The manuscript is in an important subject area but unfortunately has some weaknesses. This paper was poorly written and extensive editing of the English language and style is required.

The introduction does not provide sufficient background and relevant references. The introduction does not properly state the problem or question of the study which makes it difficult to point out the gap that the author(s) was/were trying to fill.

The research design is not appropriate and the methods were not adequately described.  

There were some errors in the presentation of the results.The Figures were not presented in the right manner as most of the text is not clear.

The “Discussion” section does not show how their study was different from previous studies that have been conducted.

I have highlighted all my comments in the attached pdf file.

This paper was poorly written and extensive editing of the English language and style is required.

Reviewer 5 Report

pls refer an attachment

Round 2

Reviewer 2 Report

This manuscript still requires minor revisions:

Q1: “double carbon goal” is an oral expression. I suggest the authors to enhance the readability of the manuscript with officially defined names. For example, carbon peak and carbon neutrality.

Q2: Can you explain more about the reason for choosing these 5 variables in equation 3? These reasons need to be elaborated by citing relevant literatures. 

Q3:

In section 3.4, it is necessary to explain the principle of cohesive subgroup analysis here. There are many different ways to analyze the network. Why to use this method to implement the subgroup analysis? 

Reviewer 3 Report

Thank you very much

Reviewer 4 Report

Dear authors

You did not correct most of the things I suggested. I have highlighted them again in the attached pdf document. 

Extensive editing of English language is required
